# Plant beta-diversity across biomes captured by imaging spectroscopy

Anna K. Schweiger [1,2 ✉] & Etienne Laliberté [1]

Monitoring the rapid and extensive changes in plant species distributions occurring worldwide requires large-scale, continuous and repeated biodiversity assessments. Imaging spectrometers are at the core of novel spaceborne sensor fleets designed for this task, but the degree to which they can capture plant species composition and diversity across ecosystems has yet to be determined. Here we use imaging spectroscopy and vegetation data collected by the National Ecological Observatory Network (NEON) to show that at the landscape level, spectral beta-diversity—calculated directly from spectral images—captures changes in plant species composition across all major biomes in the United States ranging from arctic tundra to tropical forests. At the local level, however, the relationship between spectral alpha- and plant alpha-diversity was positive only at sites with high canopy density and large plant-to-pixel size. Our study demonstrates that changes in plant species composition and diversity can be effectively and reliably assessed with imaging spectroscopy across terrestrial ecosystems at the beta-diversity scale—the spatial scale of spaceborne missions—paving the way for close-to-real-time biodiversity monitoring at the planetary level.

---

[1] Institut de recherche en biologie végétale, Département de sciences biologiques, Université de Montréal, Montréal, QC H1X 2B2, Canada. [2] Remote Sensing Laboratories, Department of Geography, University of Zurich, 8057 Zurich, Switzerland. ✉email: anna.k.schweiger@gmail.com

Global change, including climate and land use change, is altering the distribution of plant species worldwide. Over the last half-century, the rate of global biodiversity loss has continuously exceeded that incurred during the Holocene, as human impacts on our planet have escalated[1]. In addition, increasingly rapid range shifts are leading to changes in plant species composition and the loss of local plant diversity, putting the sustainability of current management practices at risk[2]. However, our ability to effectively monitor those changes using traditional field surveys is limited by logistical and financial constraints. As a result, large portions of the globe, particularly in the global south, remain understudied[3]. The scientific community has recognized the need for global, long-term and spatially complete biodiversity data to guide conservation actions, and this requires remote sensing[3–5]. Space agencies worldwide are investing in novel sensors for assessing and monitoring the composition, structure and health of terrestrial ecosystems[6–8], and imaging spectrometers (providing hyperspectral data with hundreds of spectral bands) are at the core of these sensor fleets. For example, the National Aeronautics and Space Administration's (NASA) Surface Biology and Geology (SBG) mission will acquire global spectroscopic visible to shortwave infrared (VSWIR; 380–2500 nm) imagery at high spatial resolution (~30 m × 30 m pixel size) and sub-monthly temporal resolution[7], with one of its main priorities being the quantification of vegetation distribution and composition (https://sbg.jpl.nasa.gov/satm). Similarly, the European Space Agency (ESA) is preparing its Copernicus Hyperspectral Imaging Mission for the Environment (CHIME) to support biodiversity management[9]. In addition, data collected by the Italian Space Agency's Precursore Iperspettrale della Missione Applicativa (PRISMA)[10] are already available for access (https://prismauserregistration.asi.it/).

Imaging spectroscopy is the most promising remote sensing approach to map the taxonomic and functional diversity of vegetation[11], but translating spectra to species or plant traits requires extensive field data for calibration, which are and will remain unavailable for most of the world. Over the past years, spectral diversity—which can be calculated directly from image data—has been increasingly recognized as a valuable indicator for plant diversity that integrates phylogenetic, functional, and structural facets of canopy heterogeneity[12–14]. Until now, the potential for spectral diversity to assess plant diversity has been discussed theoretically[15,16] or explored at single sites[13,17–20]. However, it is still unknown if spectral diversity can capture spatial variation in plant species composition and diversity to a sensible degree across biomes. If it did, then spectral diversity could become a globally relevant remote sensing data product to monitor the rapidly changing state of the world's plant diversity.

Here, we used spectral images and vegetation inventories collected by the National Ecological Observatory Network (NEON) to determine the degree to which spectral diversity captures plant species composition and diversity across the United States (Fig. 1a). NEON sites cover a diverse range of ecosystems (Fig. 1b, Supplementary Table 1), providing a unique opportunity to evaluate spectral diversity as an indicator for plant diversity across biomes ranging from arctic tundra to tropical forests. In addition, spectral and vegetation data are collected in a standardized way across all sites and at a spatial scale that is relevant for future satellite missions (20 m × 20 m plot size for vegetation surveys). Further, the NEON spectral dataset, given its high spatial resolution (1 m × 1 m pixel size), allowed us to explore links between spectral and plant diversity at different spatial scales (Supplementary Fig. 1). At the local scale, the diversity within plant communities (or research plots) is referred to as alpha-diversity. At the regional or landscape scale, the spatial variation in species composition among communities is referred to as beta-diversity. For investigating the spectral-diversity—plant-diversity relationship at the alpha-scale, we used plot-level species inventories (alpha-diversity metrics calculated from percent cover per species and plot) and pixel-level spectral data (spectral variance among the 1 m × 1 m image pixels per plot[15]). For investigating the spectral-diversity—plant-diversity relationship at the beta-scale, we used the same species inventories (beta-diversity metrics calculated from percent cover per species and plot) but plot-level spectral data (spectral variance among the mean spectra of 20 m × 20 m research plots).

Simulations have shown that plant alpha-diversity metrics, such as the number of species per community, can be best estimated from spectral imagery when the size of image pixels roughly equals the size of individual plants[21,22]. While we expect that the NEON airborne spectral data is of sufficiently high spatial resolution to express alpha-diversity at sites dominated by forest, alpha-diversity should be less traceable at sites dominated by herbaceous vegetation, since at these sites pixel sizes will be far larger than individual plants, approximating the size of plant communities. Beta-diversity is a fundamentally important facet of biodiversity that is strongly affected by global environmental change[23,24] and directly relevant for future satellite missions because it is, by definition, a measure derived from community-level information[25]. However, measuring beta-diversity or compositional turn-over remotely has received less attention than alpha-diversity[26]. Although links between spectral beta- and

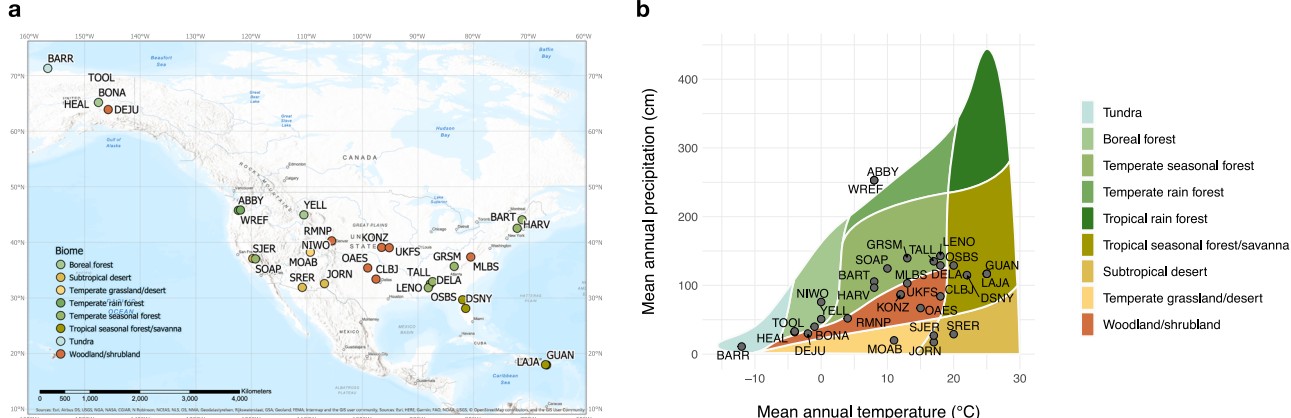

**Fig. 1 Location of NEON sites.** The NEON sites used in this study are located (**a**) across the entire United States and **b** cover all major biomes except for tropical rainforest. Colors represent different biomes; for site abbreviations and characteristics see Supplementary Table 1.

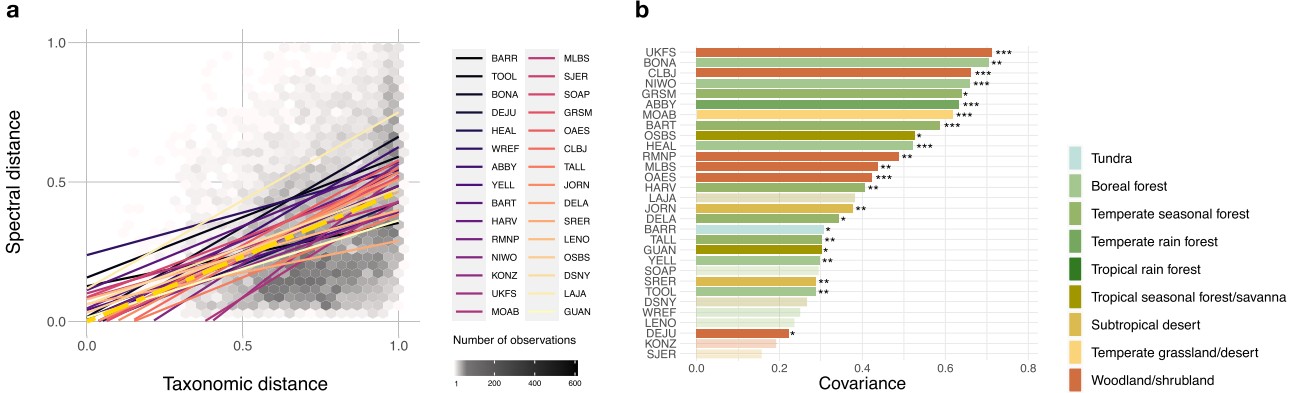

**Fig. 2 Spectral variation among plant communities captures differences in species composition. a** At each site, the average pairwise spectral distance among plots increases with their average pairwise taxonomic distance. Sites are ordered across a latitudinal gradient from north (dark colors) to south (light colors). The overall relationship between pairwise spectral and taxonomic distances across all NEON sites is displayed in golden color and dashed line ($n = 13222$, $r^2 = 0.18$, $b = 0.47$, $t_{13221} = 191.2$, $P < 0.001$). Significance of the relationship between spectral and taxonomic distance was assessed using two-sided t-tests; the number of observations is indicated by the gray tiles in the background. **b** Covariance between plot-wise ordinations of mean spectra and plant species inventories per site. Colors represent different biomes and stars significance levels, \*\*\*$P \leq 0.001$, \*\*$P \leq 0.01$, \*$P \leq 0.05$, no star and transparent shading indicates $P > 0.05$ (not significant). For site abbreviations see Supplementary Table 1, for statistics see Supplementary Tables 2, 3, for results per site see Supplementary Fig. 2.

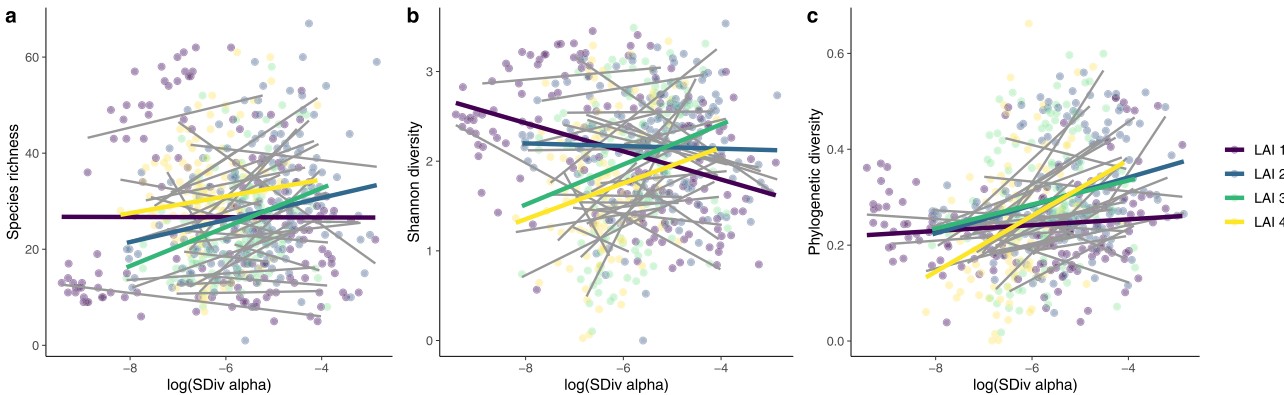

**Fig. 3 Spectral-diversity–plant-diversity relationships at the alpha-scale depend on leaf area index (LAI).** Relationships between spectral alpha-diversity and **a** plant species richness (LAI 1: $n = 157$, $P = NS$; LAI 2: $n = 165$, $r^2 = 0.08$, $b = 3.23$, $t_{163} = 3.96$, $P < 0.001$; LAI 3: $n = 149$, $r^2 = 0.05$, $b = 3.10$, $t_{147} = 3.05$, $P = 0.003$; LAI 4: $n = 118$, $r^2 = 0.02$, $b = 2.65$, $t_{116} = 1.90$, $P = 0.06$), **b** Shannon index (LAI 1: $n = 157$, $r^2 = 0.16$, $b = -0.17$, $t_{155} = -5.54$, $P < 0.001$; LAI 2: $n = 165$, $P = NS$; LAI 3: $n = 149$, $r^2 = 0.05$, $b = 0.21$, $t_{147} = 2.85$, $P = 0.005$; LAI 4: $n = 118$, $r^2 = 0.05$, $b = 0.25$, $t_{116} = 2.70$, $P = 0.008$), and **c** phylogenetic diversity (LAI 1: $n = 157$, $P = NS$; LAI 2: $n = 164$, $r^2 = 0.05$, $b = 0.02$, $t_{162} = 3.21$, $P = 0.002$; LAI 3: $n = 149$, $r^2 = 0.04$, $b = 0.03$, $t_{147} = 2.60$, $P = 0.01$; LAI 4: $n = 118$, $r^2 = 0.12$, $b = 0.06$, $t_{116} = 4.13$, $P < 0.001$). As phylogenetic diversity measure we used phylogenetic species evenness (PSE). Significance was assessed using two-sided t-tests; colors represent four classes of LAI: LAI 1 = [0.13, 0.634], LAI 2 = (0.634, 1.183], LAI 3 = (1.183, 1.84], LAI 4 = (1.84, 3.80]). For spectral-diversity–plant-diversity relationships at the alpha-scale per site see Supplementary Fig. 3.

taxonomic beta-diversity have been shown in some tropical forests[19,20], the extent to which they hold across other ecosystems remains to be investigated.

In this work, we provide evidence for a consistently positive relationship between spectral beta- and plant beta-diversity across ecosystems. This demonstrates that changes in plant species composition can be tracked from arctic tundra to tropical forests at the spatial resolution of spaceborne imaging spectrometers and without the need for extensive field data.

## Results

We found that across all NEON sites, spectral dissimilarity among plots measuring 20 m × 20 m increased with their dissimilarity in plant species composition–a measure of plant beta-diversity (Fig. 2a, Supplementary Table 2, Supplementary Fig. 2). Testing the degree of correspondence between ordinations of

mean plot-level spectra and plant inventories revealed significant covariance for 23 out of the 30 NEON sites with on average 47% of the total variation in plant inventories explained by spectra (Fig. 2b, Supplementary Table 3).

At the alpha-scale, on the other hand, we found that the spectral-diversity–plant-diversity relationship depended on environmental characteristics, including canopy density (Fig. 3, Supplementary Tables 4–6). Overall, sites with closed canopy cover—a leaf area index (LAI) greater than ~1—showed stronger positive relationships between spectral and plant alpha-diversity than sites with more open vegetation (Fig. 3). Interestingly, when accounting for environmental differences among sites, spectral alpha-diversity was a significant predictor for local plant diversity in linear mixed-effects models (Fig. 4, Supplementary Tables 4–6). This was the case even though our models, which predicted local plant diversity with high accuracies (species richness: coefficient of determination ($r^2$) = 0.66, regression

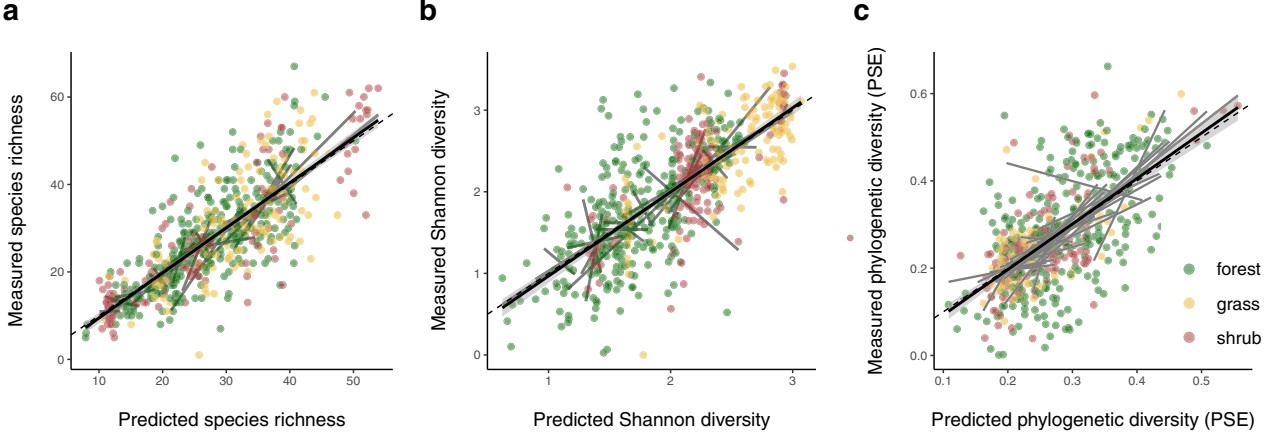

**Fig. 4 Spectral alpha-diversity predicts plant alpha-diversity.** Correlations between predicted and measured **a** plant species richness ($n = 589$, $r^2 = 0.66$, $b = 1.03$, $t_{587} = 33.80$, $P < 0.001$), **b** Shannon index ($n = 589$, $r^2 = 0.57$, $b = 1.03$, $t_{587} = 27.72$, $P < 0.001$), and **c** phylogenetic diversity ($n = 588$, $r^2 = 0.40$, $b = 1.04$, $t_{586} = 19.87$, $P < 0.001$) per plot calculated from mixed effects models including spectral alpha-diversity, vegetation type (forest, grassland, shrubland), mean leaf area index, latitude, elevation, precipitation and temperature. As phylogenetic diversity measure we used phylogenetic species evenness (PSE). The black line shows the overall model fit, one standard deviation is shaded in gray, the dashed line is the 1:1-line, gray lines are the linear regressions per site, dot colors represent vegetation types. Significance was assessed using two-sided t-tests, no adjustments for multiple comparisons were made; for mixed effects model statistics see Supplementary Tables 4–6.

coefficient ($b$) = 1.03, $t_{587} = 33.80$, $P < 0.001$; Shannon index: $r^2 = 0.57$, $b = 1.03$, $t_{587} = 27.72$, $P < 0.001$; phylogenetic diversity: $r^2 = 0.40$, $b = 1.04$, $t_{586} = 19.87$, $P < 0.001$), contained well known factors explaining global biodiversity patterns, including latitude, elevation, mean annual temperature and precipitation. Together, these results suggest that spectral alpha-diversity as measured from NEON spectral imagery might act as an indicator of plant alpha-diversity in high LAI-ecosystems containing large plants (e.g., temperate forests), but not in low LAI-ecosystems with small plants (e.g., tundra).

## Discussion

Our results show that spatial variation in plot-level reflectance captures changes in plant species composition (i.e., beta-diversity) at the proposed scale of upcoming satellite missions (~ 30 m × 30 m pixel size). Importantly, this relationship holds across biomes, including ecosystems dominated by forests, shrub- and grassland (Fig. 2, Supplementary Table 3). This means that the degree of dissimilarity in spectral reflectance is directly related to plant beta-diversity or the diversity among plant communities at the regional or landscape scale, regardless of ecosystem type. Plots that are spectrally rare or distant from the average plot within a region might indicate floristically unique areas that may be of high conservation value[11,15]. Tracking changes in spectral beta-diversity over time could provide an early warning system of ecosystem change, as changes in species composition precede more severe shifts in environmental conditions. It would also allow the detection of biotic homogenization, or the loss of beta-diversity within regions[23]. In addition to biodiversity monitoring and change detection, the spectral-diversity–plant-diversity relationship at the beta-scale allows the discovery and assessment of environmental gradients that are important for plant community assembly and biogeochemical cycles[27].

In contrast to beta-diversity, the strength of the relationship between spectral alpha- and plant alpha-diversity depended on local environmental characteristics, including pixel-to-plant size ratio and LAI (Fig. 3, Supplementary Tables 4–6). In the case of the NEON imagery, with 1 m × 1 m pixels, spectral alpha-diversity predicted plant alpha-diversity best in forests with closed canopies (LAI ≥ 1) consisting of mature trees (crown

diameter ≥ 2 m, Supplementary Fig. 3). These ecosystems might also be better suited for spectral analysis of plant alpha-diversity because edge and shadow effects are likely less prevalent than in ecosystems with more open canopies, at least at the 1 m × 1 m pixel size that the NEON imagery provides. Notably, the relationships between spectral and phylogenetic diversity metrics were stronger than between spectral and taxonomic diversity metrics (species richness and Shannon index; Fig. 3). This is likely because spectral dissimilarity among species depends on their functional dissimilarity and evolutionary divergence time[13], which are captured by phylogenetic diversity metrics but not taxonomic ones[28].

Our results demonstrate that the spatial resolution of upcoming spaceborne imaging spectrometers allows monitoring changes in plant species composition at the community or beta-diversity scale across a range of ecosystem types that encompass all major terrestrial biomes. Our study did not include tropical rainforests, but extends earlier work conducted in that biome where this relationship had been found[19,20]. While spaceborne sensors are probably less well suited to assess and monitor changes in plant alpha-diversity in some ecosystems because of the mismatch of plant-to-pixel size, this gap can be filled by spectrometers operated from airplanes or unoccupied aerial vehicles (UAVs) providing spatial resolutions at the m- and cm-scale, respectively. For getting the most out of spectroscopic methods, integrated approaches to remote sensing of plant diversity that combine the strengths of field work, UAVs, airplanes and satellites are needed[3,5,29,30]. Such integrated approaches could be based on satellite spectroscopy for detecting at the landscape scale spectrally rare areas across space and changes in plant species composition across time; before assessing these areas of interest in detail with airborne remote sensing and, whenever possible, field data.

Networks of sites that collect remote sensing and field data in a standardized way, such as NEON, are already set up as testing grounds for integrated approaches to the remote sensing of biodiversity, for instance by allowing airborne image acquisition close to satellite overpass times. Still, for the overwhelming portion of the Earth, we have no data on plant species on the ground and it is unlikely that we ever will. Even with sufficient funds, keeping track of the rapid, global reshuffling of plant species

distributions caused by climate and land-use change, over-exploitation, pollution and invasive species[1] will remain impossible with traditional field methods alone. Biodiversity observatories also cannot be launched everywhere. But given limited resources and the fact that we need high-quality field data to make sense of processes occurring on the ground, imaging spectroscopy can be used as a guide to direct fieldwork to key locations[15]. As we are entering a new era of imaging spectroscopy from space, our results provide strong evidence that differences and changes in plant species composition can be effectively and directly assessed across biomes via spectral diversity, paving the way for comprehensive, close-to-real time biodiversity monitoring at the planetary scale.

## Methods

**Spectral diversity**. For spectral diversity calculations, we used NEON Airborne Observation Platform (AOP) data (the orthorectified surface directional reflectance mosaic data product, DP1.30006.001[31]) collected in 2018 and covering 30 sites (Supplementary Table 1). Each mosaic file covers an area of 1 km × 1 km with a spatial resolution of 1 m. Spectral data comprised 426 bands, spanning the spectral region from ~380 nm to ~2510 nm at 5 nm band spacing. Image processing is done by NEON and based on the industry standard ATCOR[32] without correcting for BRDF (bidirectional reflectance distribution function) effects. Details can be found by following the dataset's DOI[31] and scrolling down to the documents under Collection and Processing—Documentation. In addition, we masked atmospheric water absorption bands from 1340 to 1445 nm and 1790 to 1955 nm, and removed spectral bands at the beginning and end of the measured spectrum, i.e., ≤400 nm and ≥2400 nm. We applied an NDVI (normalized difference vegetation index)-mask[33] to exclude non-photosynthetically active vegetation and a near-infrared (NIR) shade-mask[34] calculated as mean reflectance (ranging from 0–1) between 752 nm and 1048 nm. We used site-specific NDVI- and vegetation type-specific NIR shade-mask thresholds due to the large variation in vegetation cover and structure across our sites (Supplementary Table 1). For sites with sparse vegetation cover, such as desert sites, we used a lower greenness-threshold (e.g., NDVI ≥ 0.2 at SRER) compared to sites with high vegetation cover, such as temperate forests (e.g., NDVI ≥ 0.8 at HARV). As NIR shade-mask thresholds we chose mean NIR reflectance ≥ 0.18 for plots dominated by forest, mean NIR reflectance ≥ 0.2 for plots dominated by shrubland and mean NIR reflectance ≥ 0.22 for plots dominated by grassland. At least 50% of all pixels within each plot needed to pass the NDVI- and NIR shade-mask thresholds for the plot to be included in further analyses. We checked the validity of our thresholds using NEON's high resolution camera imagery (DP1.30010.001)[35] with 10 cm spatial resolution, collected simultaneously with the spectral data.

We selected spectral alpha-diversity ($SD_\alpha$)[15] over other existing spectral diversity metrics because it measures spectral variance, which we felt more closely reflected abundance-weighted taxonomic diversity measures. As plant communities, we used NEON's distributed base plots measuring 20 m x 20 m within which plant inventories are conducted. For each site, we extracted spectral reflectance values per plot and brightness-normalized all spectra to account for illumination differences[36]. We performed a PCA with type I-scaling to reduce data dimensionality[15] and selected PCs until more than 95% of the total spectral variation was explained. We calculated $SD_\alpha$ as the sum of the squared deviations of every pixel and spectral feature (PC) per community from the mean spectral feature of that community standardized by the number of pixels in the community[15].

**Plant diversity**. We calculated taxonomic diversity metrics using two datasets: i) percent cover of herbaceous and understory plants (i.e., plants <3 m) from NEON's plant presence and percent cover data (DP1.10058.001)[37] and ii) tree inventories, including identity, location, crown diameter and height, from NEON's woody plant vegetation structure data (DP1.10098.001)[38]. Details regarding the sampling design for generating these datasets can be found by following the datasets' DOIs[37,38] and scrolling down to the documents under Collection and Processing - Documentation. Since not all NEON sites were inventoried in 2018, the year of spectral image collection, we used plant inventories collected between 2016 and 2019. Most plots and sites used in our study contained individuals ≥3 m (trees, tall shrubs). However, for individuals ≥3 m NEON's plant presence and percent cover (DP1.10058.001)[37] data does not report crown cover, but only basal diameter (when there is no vegetative growth <3 m) or percent cover of foliage along the stem <3 m (when there is vegetative growth <3 m). For matching spectral data and plant inventories it is, however, important to scale inventories to vegetation cover as seen from above. We thus mapped all trees recorded in the woody plant vegetation structure[38] data based on their location, crown diameter and height[39] (Supplementary Fig. 4). For each plot, we calculated crown cover per tree as seen from above taking overlapping crowns into account by ranking the trees according to their height. We then scaled plant cover <3 m to the area per plot not covered by tree crowns and combined both, tree cover and scaled herbaceous/understory inventories. We calculated taxonomic alpha-diversity metrics, i.e., species richness and Shannon index using the R[40] package vegan[41]. For calculating phylogenetic alpha-diversity, we generated 100 phylogenies with V.phylomaker[42] in R using the phylogeny released by ref. [43] as the backbone mega-tree, with new tips bound to randomly selected nodes at and below the genus- or family-level basal node (scenario 2, 100 repeats in V.phylomaker). We used the taxonomic name resolution service (http://tnrs.iplantcollaborative.org/) to match species names to the backbone phylogeny; and we attached missing species, unknown species and plant genera with uncertain identification to the phylogenies using close species– or genus–level relatives. Then, we calculated phylogenetic species evenness (PSE) per community based on vegetation cover seen from above and the 100 phylogenies using the R package picante[44], and averaged the result. Initially, we also intended to calculate functional diversity metrics calculated from the plant foliar physical and chemical properties data collected by NEON[45]. However, at the time of our analysis this dataset was too small to derive any meaningful metrics across all 30 sites. We calculated taxonomic beta-diversity per site using Hellinger distances and the R script provided in ref. [46].

**Statistical analysis**. We assessed the association between spectral and taxonomic variation within sites (i.e., at the beta-diversity scale) based on spectral and taxonomic distance among plots. For each plot within a site, we calculated the distance to all other plots at the site based on the plot-level average spectrum and vegetation cover as seen from above, using Euclidean and Hellinger distances for spectral and plant inventory data, respectively. To assessed the overall association between spectral variation and plant community composition, we fit linear regression models between spectral and taxonomic distances per plot, per site and across all sites. In addition, we assessed the correspondence between ordinations of plot-wise mean spectra and plant inventories with co-inertia analysis using Monte Carlo testing on the sum of eigenvalues with 999 permutations as implemented in the R package ade4[47].

To assess the potential dependence of the relationship between spectral alpha- and plant alpha-diversity on site characteristics, we used mixed effect models as implemented in the R package nlme[48] with site identity as the random effect. Environmental variables included in the models were mean LAI per site calculated from NEON's spectrometer LAI mosaic data[49] and site characteristics listed by NEON, i.e., main vegetation type (forest, grassland, shrubland), latitude, elevation, mean annual temperature and mean annual precipitation (Supplementary Table 1).

**Reporting summary**. Further information on research design is available in the Nature Research Reporting Summary linked to this article.

## Data availability

All data used in this analysis are available from NEON: https://doi.org/10.48443/qeae-3×15, https://doi.org/10.48443/4e85-cr14, https://doi.org/10.48443/abge-r811, https://doi.org/10.48443/e3qn-xw47, https://doi.org/10.48443/h2rb-pj34.

## Code availability

The R code is available on GitHub at https://github.com/elaliberte/specdiv (https://doi.org/10.5281/zenodo.6385476) and https://github.com/annakat/NEON_crown_area (https://doi.org/10.5281/zenodo.6383923).

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

## Acknowledgements
The authors thank Dave Barnett, Tristan Goulden, Claire Lunch, Courtney Meier, Kate Thibault, Samantha Weintraub-Leff and the entire NEON team for their support. We also thank Maarten B. Eppinga and Matthew Kaproth for providing thoughtful feedback on our manuscript. We thank CABO members for stimulating discussions about our study. A.K.S. acknowledges support by the University Research Priority Program Global Change and Biodiversity of the University of Zurich. This study was funded by Discovery Grants from the Natural Sciences and Engineering Research Council of Canada (NSERC; RGPIN-2014-06106, RGPIN-2019-04537 to E.L.) and a Discovery Frontiers Grant to support the Canadian Airborne Biodiversity Observatory (CABO) from NSERC (509190-2017 to E.L.). A.K.S. dedicates this work to the memory of her late father, Dr. Heinz Schweiger.

## Author contributions
A.K.S. and E.L. conceptualized the ideas and developed the methodology for this work. A.K.S. lead data analyses and visualization. A.K.S. and E.L. wrote and revised the manuscript together.

## Competing interests
The authors declare no competing interests.
