## [Peer Review File · Nature Communications]

Reviewer comments, first round

Reviewer #1 (Remarks to the Author):

In this paper, the authors present a large and powerful dataset exploring the ability of airborne imaging spectroscopy data to measure fundamental properties of plant biodiversity (alpha and beta diversity) across biomes in the USA. The authors present some mixed results, but overall, I think this is an excellent paper that addresses a timely and vitally important issue. The main result that imaging spectroscopy can be used to estimate beta diversity across and within ecosystems is an important finding that has not been shown anywhere else. The paper is concise, broadly clear and well written and I would recommend it for publication in Nature Communications. The authors should be commended for their excellent work, bringing together data from across very different plant communities and presenting a general framework that I think represents a major step forward.

That being said, I have a number of comments that I think could help improve the manuscript. I hope the authors find them useful! I've broken comments down into a few different sections. If anything is unclear I'm happy to answer questions directly.

Best wishes,
Freddie Draper

Alpha diversity results

In general, I think the authors are overplaying a bit the strength of the alpha diversity results, e.g. paragraph starting on line 93. Basically, the relationship between spectral alpha diversity and taxonomic/phylogenetic alpha diversity is very weak across sites, within individual sites it seems extremely variable, sometimes it works well sometimes it doesn't, sometimes it's even a negative correlation. The model that includes all the environmental variables works pretty well, but I guess the model performance would be pretty similar without the spectral diversity info, i.e. we already know species richness is strongly related to climate, latitude etc. Does spectral diversity bring anything to the party in this context?

I think the authors could be a bit more up front about this and discuss the reasons why that might be. They do discuss the mismatch between plant size and pixel size, which I agree is probably the key point, but there are more issues than just this. They could also suggest some ways forward on this front. Some ideas:

- The approach for quantifying spectral alpha. The authors use one approach, there are more out there e.g. calculating alpha diversity based on unsupervised classifications (spectral species idea). For the record, I think the authors approach is better, to me spectral species idea doesn't make much sense for alpha. Nevertheless, I think the point stands that we don't really know the best way to estimate spectral alpha diversity, and that should be a priority for future research.
- The approach for quantifying taxonomic diversity. To me these methods were a bit weird. I know they are covered in depth by other publications, but a couple of things stood out. The data used was presence and percent cover, which was converted to veg cover "as seen from above", this is an interesting approach, and I think that it makes sense, but it's also making some assumptions. But I'd

be interested to know how the results changed if straight species richness was used.

- The use of drones with 1 x 1 cm pixels would get rid of the number of plants per pixel issue for most ecosystems. Maybe an opportunity to advocate for an approach that integrates across scales and resolutions.
- The authors discuss this mismatch between pixel and plant, but I don't think they talk about the differences among sites in terms of individuals per plot. A 20 x 20 m plot will contain tens to maybe hundreds of individual plants in open or shrubby communities, but in a tropical forest, a single tree crown could cover the entire plot, or could contain 20 individuals belonging to 20 species. Basically, the 20x 20 m plots will be under sampling canopy tree species richness in tropical forests, its actually a real shame that larger plots weren't used, but there we are.

This variation in individuals per plot shouldn't be an issue if the spectral data has been collected for exactly the same 20 x 20 m area, meaning the plot GPS coordinates and the ortho mosaic are perfectly located and aligned. In my experience this is a big challenge! Especially in forested regions. Even if they are offset by a few meters it could lead to big errors especially in diverse forests with large crowns.

An alternative approach would be to instead of trying to hit exactly the same 20x 20 m plot, look to characterize alpha diversity across a broader area in the spectral data and then compare this with the means of the plots in that broad location. Less precise, but I the relationship would be more robust to errors in georeferencing. I'm not really suggesting the authors do this, but I think it would be helpful to flag that their approach doesn't seem to work well, and there is scope for further work in this direction.

Beta diversity results

I agree with the authors that these results provide compelling evidence for the capabilities of spectral data to estimate beta diversity with and among ecosystems. This is a big and important result.

I think the authors are a little bit guilty of downplaying the importance of field data. They do touch on this in the final paragraph, but I think they could go further. While spectral data will hopefully tell us where and when there are changes in plant species composition, or where there are particularly interesting or unusual plant communities, e.g. in the case of unexplored tropical forests. It probably won't be able to tell us exactly what those changes are in terms of species. It might be helpful then to include a sentence about how spectroscopy could be used to guide and direct fieldwork to key locations, to optimize limited resources. I think it's a bad idea to set up imaging spectroscopy as an alternative to fieldwork, we need both, and we need them to be integrated.

The authors present the results in terms of variation in Hellinger distances for spectral data and species data. Have they tried doing some kind of ordination with these distances (e.g. NMDS), extracting the axis scores and then using them for comparison? We have done this previously and found that the ordination steps helps to isolate floristic gradients within these multivariate compositional datasets. Mainly just curious here.

General comments

The authors frame the paper around the several upcoming and exciting spectroscopy satellite platforms. As the authors mention, these will be at a very different spatial resolution, more 20 x 20 m pixels rather than 1x 1 in this study. At this scale, all ecosystems will suffer to some extent with the issue of multiple individuals in a single pixel. Will this massively impact their ability to capture plant diversity? The authors could resample the 1x1 data to 20x20 and test what the impacts are on calculating diversity metrics directly. Maybe something for the next paper.

The NEON dataset doesn't include any true tropical rainforest sites. Tropical forests are really key for biodiversity, but present perhaps a different set of challenges due to exceptionally high species richness. Might be good to acknowledge this gap and the challenges might hold.

I think a few more details are needed in the methods, especially surrounding the methods used to calculate spectral diversity. I know the author has published these previously, but I think a quick recap of the general procedure would be helpful to readers.

Reviewer #2 (Remarks to the Author):

Title

Plant biodiversity across biomes 1 captured by spectroscopy

Manuscript # NCOMMS-22-52204-T

The goal of the paper is

The present study investigates very well the extent to which imaging spectroscopy can assess the composition and diversity of plant species in different ecosystems based on the NEON dataset. The study shows that changes in plant species composition and diversity can be effectively and reliably assessed using imaging spectroscopy in terrestrial ecosystems and at a spatial scale relevant to space-based missions.

The paper is excellently written in a clear and understandable manner. Only a few methodological approaches should be described in more detail.

The paper should be accepted, if the following comments are discussed in more detail.

Abstract:

The results of the study are formulated too generally in the abstract. Please revise and flesh out the results of this study in the abstract.

Detailed comments:

Line 99-100

You describe that the strongest positive relationship between spectral and plant alpha diversity in an open vegetation is enhanced by the inclusion of environmental factors such as (elevation, temperature or precipitation).

It might be worthwhile to include other geovariates or geohydrological variables in the modelling to achieve model improvement. Global and comparable data are available for this purpose.

Amatulli, G., McInerney, D., Sethi, T., Strobl, P., Domisch, S., 2020. Geomorpho90m, empirical evaluation and accuracy assessment of global high-resolution geomorphometric layers. *Sci. Data* 7, 1–18. <https://doi.org/10.1038/s41597-020-0479-6>

Methods:

You constantly describe that you use imaging spectrometer data from NEON from airborne campaigns. However, you also mention airborne images in lines 252-255. Please provide more detailed information on the imaging spectrometer data (camera type, number of wavelengths, recording times).

Likewise, what corrections were made to correct for the temporal variability of possibly different acquisition times?

In your study you use many different vegetation types. The spectral NEON data are available with a spatial resolution of 1x1m. Please mention whether a BRDF correction of the spectral data was carried out and what influence this has or could have on your model results.

Your study did not describe how the LAI was calculated. Please explain this again.

Further recommendations:

As you discuss the issue of transferability and scaling of imaging spectrometer RS data, I do not understand why you do not include available spaceborne hyperspectral RS data in your analysis such as DESIS or PRISMA?

Reviewer #3 (Remarks to the Author):

Regarding the manuscript titled "Plant biodiversity across biomes captured by spectroscopy":

I recommend that this manuscript be published in this journal for the following reasons:

1. The findings that spectral diversity tracks taxonomic biodiversity, both alpha and beta, are extremely important.
2. Given the wide range of environmental conditions in the NEON sites, these results confirm expectation regarding expression of biodiversity in spectral signals better than has previously been done. As such, these results will serve well to lend credence to the explosion of spectroscopy-based diversity assessments that will take place as several satellites launch in this decade.
3. I also appreciate that the article is well-written (the Methods can use a little work, more on this below) and easy to follow, and the standardization in the NEON network of study sites leaves little room for complaint as far as soundness of the results.

I have very few comments about the main text...

Line 89 - "plot-level spectra" is maybe a little too vague even if this is described later in the Methods, can you elaborate just a little to help the read understand.

Line 122 - Similarly for "spectral alpha-diversity metrics", I think it would help if you briefly describe such a metric here.

The methods section was a little less clearly written, and it left me with a few more questions, though nothing that can't be fixed with some minor edits...

General

- A really helpful addition to the Methods would be a brief description of the plot structure of a NEON site. I'm not entirely sure what the difference between a "inventory" plot and a "focal" plot.
- There are a few misspelled words (or rather correctly spelled, but wrong endings like "removed" instead of "remove" L261) and fairly inconsistent use of commas in this section that made it so I had to reread a few passages. It would be good to scan over this again fixing the words and perhaps breaking up long sentences to reduce comma use.

Line 255 - Good to state the spectral resolution here, 5 nm band spacing?

Line 261 - Can you describe the NIR shade mask a bit more, starting with which band(s) and how are they combined?

Line 266 - Is NIR > 0.18 here 18% reflectance?

Line 268 - It sounds like the PCA was done across all spectra across all sites as well, and not by plot or some other subgrouping. Maybe just specify here if this is true.

Line 270 - Can you briefly describe the spectral diversity methodology of Laliberté et al. 2020 here? It is fairly important to the results, and I don't think a reader should have to look this up.

Line 284 - I'm not sure what the last clause of this sentence means. Maybe explain how shrub data are measured differently and why such a scaling needs to be done.

Reviewer #1 (Remarks to the Author):

In this paper, the authors present a large and powerful dataset exploring the ability of airborne imaging spectroscopy data to measure fundamental properties of plant biodiversity (alpha and beta diversity) across biomes in the USA. The authors present some mixed results, but overall, I think this is an excellent paper that addresses a timely and vitally important issue. The main result that imaging spectroscopy can be used to estimate beta diversity across and within ecosystems is an important finding that has not been shown anywhere else.

The paper is concise, broadly clear and well written and I would recommend it for publication in Nature Communications. The authors should be commended for their excellent work, bringing together data from across very different plant communities and presenting a general framework that I think represents a major step forward.

- We thank the Reviewer. We are delighted to hear this overall positive assessment of our manuscript.

That being said, I have a number of comments that I think could help improve the manuscript. I hope the authors find them useful! I've broken comments down into a few different sections. If anything is unclear, I'm happy to answer questions directly.

- We very much appreciate the Reviewer's insightful comments. We address all of them below. **Red letters** indicate sections in the manuscript that we have changed.

Alpha diversity results

In general, I think the authors are overplaying a bit the strength of the alpha diversity results, e.g., paragraph starting on line 93. Basically, the relationship between spectral alpha diversity and taxonomic/phylogenetic alpha diversity is very weak across sites, within individual sites it seems extremely variable, sometimes it works well sometimes it doesn't, sometimes it's even a negative correlation. The model that includes all the environmental variables works pretty well, but I guess the model performance would be pretty similar without the spectral diversity info, i.e., we already know species richness is strongly related to climate, latitude etc. Does spectral diversity bring anything to the party in this context?

- The spectral alpha diversity term as well as interaction terms involving it are significant in the model, so yes it does bring something to the party, even after accounting for the other variables (e.g., climate, latitude). The significant interactions with other variables (e.g., LAI) explain why the relationships between spectral and taxonomic alpha diversity vary among sites because these sites differ in LAI and other environmental variables. But we agree with the Reviewer that our presentation of the spectral alpha- and beta-diversity results need to be crystal clear. That is, at the beta-diversity scale spectral diversity predicts plant diversity across biomes (and we changed our title to reflect this key result, see citation below), while at the alpha-diversity scale the relationship is more complicated since it depends on canopy density (measured as LAI).
To clarify this, we have updated the title and abstract (also suggested by Ref #3), and other sections in our manuscript referring to alpha-diversity results.

Title: "Plant **beta**-diversity across biomes captured by imaging spectroscopy"

P1, L19: "We show that at the landscape level, spectral **beta**-diversity—calculated directly from spectral images—captures **changes** in plant species composition across ecosystems ranging from arctic tundra to

tropical forests. At the local level, **however, the spectral alpha-diversity plant alpha-diversity relationship was positive only at sites with high canopy density and large plant-to-pixel size.**”

We think the alpha-diversity result is noteworthy because it suggests that spectral alpha diversity as measured from the NEON AOP might act as an indicator of taxonomic alpha diversity in high LAI-ecosystems (e.g., temperate forests) but not in those with low LAI (e.g., shrublands, tundra). We explain this on P6, L154:

P6, L133: “In the case of the NEON imagery, with 1 m × 1 m pixels, spectral alpha-diversity **predicted plant alpha-diversity** best in forests with closed canopies (LAI ≥ 1), consisting of mature trees (crown diameter ≥ 2 m, Extended Data Fig. 3).”

The key result of our study that “spectral diversity predicts plant diversity at the beta-scale” should be much clearer now.

I think the authors could be a bit more up front about this and discuss the reasons why that might be. They do discuss the mismatch between plant size and pixel size, which I agree is probably the key point, but there are more issues than just this. They could also suggest some ways forward on this front. Some ideas:

The approach for quantifying spectral alpha. The authors use one approach, there are more out there e.g., calculating alpha diversity based on unsupervised classifications (spectral species idea). For the record, I think the authors approach is better, to me the spectral species idea doesn't make much sense for alpha. Nevertheless, I think the point stands that we don't really know the best way to estimate spectral alpha diversity, and that should be a priority for future research.

- Like the Reviewer, we also think that our approach for measuring spectral alpha diversity is most appropriate here. We are aware that other spectral alpha diversity metrics exist, including the number of spectral “species”, and we discuss those metrics and how they compare to ours in a previous paper (Laliberté et al. 2020). However, it was beyond the scope of the present study to compare spectral alpha diversity metrics. We added a sentence in the Methods to explain why we selected our metric:

P15, L299: “**We selected spectral alpha-diversity (SD α) *sensu* Laliberté, Schweiger and Legendre (2020)¹⁶ over other existing spectral alpha-diversity metrics because it measures spectral variance, which we felt more closely reflected abundance-weighted taxonomic diversity measures.**”

The approach for quantifying taxonomic diversity. To me these methods were a bit weird. I know they are covered in depth by other publications, but a couple of things stood out. The data used was presence and percent cover, which was converted to veg cover “as seen from above”, this is an interesting approach, and I think that it makes sense, but it's also making some assumptions. But I'd be interested to know how the results changed if straight species richness was used.

- The issue here is that we needed to combine two NEON datasets to quantify plant taxonomic diversity at the plot scale. Dataset 1, NEON's plant presence and percent cover data (DP1.10058.001), reports percent cover for herbaceous and understory plants < 3 m. However, for plants ≥ 3m this dataset only contains stem basal diameter (for individuals without vegetative growth < 3 m) or percent cover of foliage < 3 m (for individuals with vegetative growth < 3 m). We are pasting an illustration for this from NEON's product documentation below (“Figure 2”). Most of the study plots we used did however contain large trees > 3 m, whose crown diameters exceeded their basal diameter. Therefore, we included NEON's woody plant vegetation structure data (DP1.10098.001) as dataset 2, as it contains information on tree identity, location, height and crown diameter.

Figure 2. Estimates of cover should include all vegetative material < 300cm in height. For herbaceous growth (A), and shrubs (B) < 300cm, record the total combined cover by species; for tall trees with no woody branches or foliar growth < 300cm (C) record basal area and a height of > 300cm should be noted for that species; for trees (D) and shrubs (E) > 300cm that also have vegetative growth < 300cm, record the cover of vegetative growth < 300cm and indicate the presence of individuals > 300cm in height for that species. There will be instances when herbaceous growth <300cm (A) and trees >300cm (C) of the same species are found in the same 1m² subplot, in these cases record the combined cover and indicate the presence of individuals by species > 300cm.

We used this data to map all tree crowns per plot, calculated their % cover, and scaled the herbaceous % cover to the area not covered by trees. NEON does not provide any combined inventories for understory (< 3 m) and potential overstory (\geq 3 m) vegetation, and neither something like “summary data” containing the number of species per plot. Therefore, it is up to the user to combine inventories in a meaningful way (for some applications only the understory data might be important), and calculate appropriate diversity metrics. As the Reviewer mentions, from a remote sensing perspective, combining all plant inventories (shrubs are also contained in the < 3 m plant inventories, DP1.10058.001) to vegetation cover as seen from above makes most sense, which is why we did this. We revised the “Plant diversity” section in the methods (see quotes below) and added Extended Data Fig. 5 (below) to the Supplement to illustrate our approach better.

P15, L311: “We calculated taxonomic diversity metrics using two datasets: **i) percent cover of herbaceous and understory plants (i.e., plants < 3 m) from NEON’s plant presence and percent cover data (DP1.10058.001)³⁶ and ii) tree metrics, including identity, location, crown diameter and height, from NEON’s woody plant vegetation structure data (DP1.10098.001)³⁷.**”

P16, L318: “**Most plots and sites used in our study contained individuals \geq 3 m (trees, tall shrubs). However, for individuals \geq 3 m NEON’s plant presence and percent cover (DP1.10058.001)³⁶ data does not report crown cover, but only basal diameter cover (when there is no vegetative growth < 3 m) or percent cover of foliage along the stem < 3 m (when there is vegetative growth < 3 m). For matching spectral data and plant inventories it is, however, important to scale inventories to vegetation cover as seen from above. We thus mapped all trees recorded in the woody plant vegetation structure³⁷ data based on their location, crown diameter and height³⁸ (Extended Data Fig. 5). For each plot, we calculated crown cover per tree as seen from above taking overlapping crowns into account by ranking the trees according to their height. We then scaled plant cover < 3 m to the area per plot not covered by tree crowns and combined both, tree cover and scaled herbaceous/understory inventories.**”

Extended Data Fig. 5. Scaling plant inventories to vegetation cover as seen from above. Similar to this example of plot 1 (delineated by the blue square) at the Abby Road site, we mapped all trees per plot based on their location, height and crown diameter. We calculated the area within the plot covered by each tree species, and scaled percent cover of herbaceous and understory plants to the area within the plot not covered by trees (white area within the blue square).

The use of drones with 1 x 1 cm pixels would get rid of the number of plants per pixel issue for most ecosystems. Maybe an opportunity to advocate for an approach that integrates across scales and resolutions.

- We fully agree with the Reviewer that integrated approaches for remote sensing of biodiversity are needed. We included a statement and cite others who have advocated for this approach.

P7 L148: “While spaceborne sensors are **probably** less well suited to monitor changes in species composition at the alpha-diversity scale **in some ecosystems** because of the mismatch of plant-to-pixel size, this gap can be filled by **spectrometers operated from airplanes or unoccupied aerial vehicles (UAVs) providing spatial resolutions at the m- and cm-scale, respectively.** For getting the most out of spectroscopic methods, **integrated approaches to remote sensing of plant diversity that combine the strengths of field work, UAVs, airplanes and satellites are needed^{1,11,28,29}**.”

The authors discuss this mismatch between pixel and plant, but I don't think they talk about the differences among sites in terms of individuals per plot. A 20 x 20 m plot will contain tens to maybe hundreds of individual plants in open or shrubby communities, but in a tropical forest, a single tree crown could cover the entire plot, or could contain 20 individuals belonging to 20 species. Basically, the 20x 20 m plots will be under sampling canopy tree species richness in tropical forests, it's actually a real shame that larger plots weren't used, but there we are.

- The Reviewer is correct that plot size matters when estimating alpha-diversity, but NEON uses the same plot size at all sites so this is what had to work with. However, in our case, the issue is not as bad as the situation the Reviewer is describing where a single tree could cover an entire plot. For plots containing trees, the number of trees within the plot ranged from 1-106, with a mean of 15 trees per plot. However, for plots containing only one tree this tree covered a maximum of 28% of the total plot area; and in this particular plot the total number of species, including understory plants, was 34 species (see Fig. R1 below). Overall, in plots containing only one tree this tree covered between 0.08-28% of the total plot area (mean=5 %) while species richness ranged from 9-44 species (mean = 26 species).

Fig. R1. NEON plots containing only one tree. Tree cover (left y-axis) is indicated by black dots, the number of species per plot (right y-axis) is indicated by red triangles.

This variation in individuals per plot shouldn't be an issue if the spectral data has been collected for exactly the same 20 x 20 m area, meaning the plot GPS coordinates and the ortho mosaic are perfectly located and aligned. In my experience this is a big challenge! Especially in forested regions. Even if they are offset by a few meters it could lead to big errors especially in diverse forests with large crowns.

- For each plot, NEON provides GPS uncertainty information. The plot coordinates and metadata can be obtained from the NEON document library <https://data.neonscience.org/documents> "All_NEON_TOS_Plots_V7". Mean and maximum horizontal accuracies, defined as the estimated horizontal distance (in meters) from the given spatial location describing the smallest circle containing the actual location, are 0.13 m (mean) and 0.36 m (max), respectively. The spatial accuracy of the imaging spectroscopy data can be assumed to be max. 1 pixel, i.e., 1 m. From this we conclude that the coordinates are "accurate enough" to allow for a matching spectral data and plot level data (plant inventories).

An alternative approach would be to instead of trying to hit exactly the same 20x 20 m plot, look to characterize alpha diversity across a broader area in the spectral data and then compare this with the means of the plots in that broad location. Less precise, but I think the relationship would be more robust to errors in georeferencing. I'm not really suggesting the authors do this, but I think it would be helpful to flag that their approach doesn't seem to work well, and there is scope for further work in this direction.

- We fully agree with the Reviewer that this would be an excellent approach if plot and/or imagery spatial accuracy was an issue, which it is not here, as explained above. The NEON plot and image spatial accuracies are high enough that we can safely assume to have "hit exactly the same 20 m x 20 m plot". We realize that this is far from always being the case, but luckily it is for the NEON data.

Beta diversity results

I agree with the authors that these results provide compelling evidence for the capabilities of spectral data to estimate beta diversity with and among ecosystems. This is a big and important result.

- We thank the Reviewer and are pleased that he recognizes the importance of this result. We have modified our title to emphasize it even further:
"Plant **beta**-diversity across biomes captured by imaging spectroscopy"

I think the authors are a little bit guilty of downplaying the importance of field data. They do touch on this in the final paragraph, but I think they could go further. While spectral data will hopefully tell us where and when there are changes in plant species composition, or where there are particularly interesting or unusual plant communities, e.g., in the case of unexplored tropical forests. It probably won't be able to tell us exactly what those changes are in terms of species. It might be helpful then to include a sentence about how spectroscopy could be used to guide and direct fieldwork to key locations, to optimize limited resources. I think it's a bad idea to set up imaging spectroscopy as an alternative to fieldwork, we need both, and we need them to be integrated.

- We fully agree with the Reviewer, and never meant to imply that spectroscopy should replace field work. In fact, in our previous paper describing the spectral diversity approach we used here (Laliberté et al. 2020), we explained that “mapping spectral diversity could be used as a biodiversity “discovery tool” to design targeted field sampling campaigns”. To clarify this, we added this and some additional thoughts about data integration to the discussion.

P7 L152: “For getting the most out of spectroscopic methods, integrated approaches to remote sensing of plant diversity that combine the strengths of field work, UAVs, airplanes and satellites are needed^{1,11,28,29}. Such integrated approaches could be based on satellite spectroscopy for detecting at the landscape scale spectrally rare areas across space and changes in plant species composition across time; before assessing these areas of interest in detail with airborne remote sensing and, whenever possible, field data.”

P8 L165: “Biodiversity observatories also cannot be launched everywhere. But given limited resources and the fact that we need high quality field data to make sense of processes occurring on the ground, imaging spectroscopy could be used as a guide to direct fieldwork to key locations¹⁶.”

The authors present the results in terms of variation in Hellinger distances for spectral data and species data. Have they tried doing some kind of ordination with these distances (e.g., NMDS), extracting the axis scores and then using them for comparison? We have done this previously and found that the ordination steps help to isolate floristic gradients within these multivariate compositional datasets. Mainly just curious here.

- This study by the Reviewer (Draper et al 2019) was actually a major source of inspiration for our study. The multivariate approach we used (co-inertia analysis) is conceptually similar to that used by Draper et al (2019), but is arguably better in the sense that it considers all axes of spectral and taxonomic variation simultaneously, and not just the primary axes.

General comments

The authors frame the paper around the several upcoming and exciting spectroscopy satellite platforms. As the authors mention, these will be at a very different spatial resolution, more 20 x 20 m pixels rather than 1x 1 in this study. At this scale, all ecosystems will suffer to some extent with the issue of multiple individuals in a single pixel. Will this massively impact their ability to capture plant diversity? The authors could resample the 1x1 data to 20x20 and test what the impacts are on calculating diversity metrics directly. Maybe something for the next paper.

- The beta-diversity component considers the average spectra (centroid) of the plots (see Fig. 1 in Laliberté et al. 2020, which we paste below). So, we are in fact already doing that for beta-diversity. We clarify our approach in the introduction section:

P4 L75: “For investigating the spectral-diversity—plant-diversity relationship at the beta-scale, we used the same species inventories (beta-diversity metrics calculated from percent cover per species and plot) and plot-level spectral data (spectral variance among the mean spectra of 20 m x 20 m research plots).”

Figure 1 Partitioning plant spectral γ -diversity into additive β and α components. A region of interest is split into a number of communities of a specific size and shape (here, 20×20 m squares, representing standard forest inventory plots). Spectral γ -diversity refers to the total spectral diversity in the entire region, calculated from pixel-level reflectance. The β component corresponds to spectral diversity *among* communities, with similar colours sharing more similar spectral composition. The α component refers to spectral diversity *within* individual communities. The left-most panel is a true colour (red-green-blue, RGB) image of an area of Bartlett Experimental Forest; colours for the other panels were obtained using the reflectance of different wavelength bands (R = 779 nm, G = 639 nm, B = 2301 nm), followed by linear stretching.

The NEON dataset doesn't include any true tropical rainforest sites. Tropical forests are really key for biodiversity, but present perhaps a different set of challenges due to exceptionally high species richness. Might be good to acknowledge this gap and the challenges might hold.

- We thank the Reviewer for this important point. We acknowledge that the NEON dataset doesn't include tropical rainforests, since the Puerto Rican sites are tropical dry forests, on P7 L144 (see quote below). At the same time, we believe that thanks to the work of the CAO/GAO group, including the great work done by the Reviewer (Draper et al. 2019), there is a good amount of evidence that assessing alpha- and beta-diversity from spectra is possible in these diverse systems. We discuss this on (P4 L89):

P4 L89: "Although links between spectral beta- and taxonomic beta-diversity have been shown in some tropical forests^{18,25}, the extent to which they hold across other ecosystems remains to be investigated."

P7 L144: "Our results demonstrate that the spatial resolution of upcoming spaceborne imaging spectrometers allows monitoring changes in **plant** species composition at the community or beta-diversity scale across a range of ecosystem types that encompass all major terrestrial biomes. **Our study did not include tropical rainforests, but extends earlier work conducted in that biome where this relationship had been found^{18,19}.**"

I think a few more details are needed in the methods, especially surrounding the methods used to calculate spectral diversity. I know the author has published these previously, but I think a quick recap of the general procedure would be helpful to readers.

- We agree, and this was also pointed out by the editor and other reviewers. We have updated to methods accordingly.

P15, L298: "**Spectral diversity calculations followed the methodology published in Laliberté, Schweiger and Legendre (2020)¹⁶. We selected spectral alpha-diversity (SD_α) *sensu* Laliberté, Schweiger and Legendre (2020)¹⁶ over other existing spectral alpha-diversity metrics because it measures spectral variance, which we felt more closely reflected abundance-weighted taxonomic diversity measures. For all analyses, we used NEON's distributed base plots measuring 20 m x 20 m, as these are the plots within which plant inventories are conducted. For each site, we extracted spectral reflectance values per plot, brightness-normalized all spectra³⁵ to account for illumination differences and performed a PCA with type I-scaling to reduce data dimensionality. We selected PCs until more than 95% of the total spectral variation was explained. Spectral diversity was calculated as the total variance among the PC values of all pixels corrected by the number of pixels.**"

Reviewer #2 (Remarks to the Author):

Title

Plant biodiversity across biomes captured by spectroscopy

Manuscript # NCOMMS-22-52204-T

The present study investigates very well the extent to which imaging spectroscopy can assess the composition and diversity of plant species in different ecosystems based on the NEON dataset. The study shows that changes in plant species composition and diversity can be effectively and reliably assessed using imaging spectroscopy in terrestrial ecosystems and at a spatial scale relevant to space-based missions.

The paper is excellently written in a clear and understandable manner. Only a few methodological approaches should be described in more detail.

The paper should be accepted, if the following comments are discussed in more detail.

- We thank the Reviewer very much for this positive feedback. We address comments and suggestions below.

Abstract:

The results of the study are formulated too generally in the abstract. Please revise and flesh out the results of this study in the abstract.

- We agree, and have updated the Abstract to make alpha- and beta-diversity results clearer:

P1, L19: “We show that at the landscape level, spectral **beta**-diversity—calculated directly from spectral images—captures **changes** in plant species composition across ecosystems ranging from arctic tundra to tropical forests. At the local level, **however, the spectral alpha-diversity plant alpha-diversity relationship was positive only at sites with high canopy density and large plant-to-pixel size.**”

Detailed comments:

Line 99-100

You describe that the strongest positive relationship between spectral and plant alpha diversity in an open vegetation is enhanced by the inclusion of environmental factors such as (elevation, temperature or precipitation).

It might be worthwhile to include other geovariates or geohydrological variables in the modelling to achieve model improvement. Global and comparable data are available for this purpose.

Amatulli, G., McInerney, D., Sethi, T., Strobl, P., Domisch, S., 2020. Geomorpho90m, empirical evaluation and accuracy assessment of global high-resolution geomorphometric layers. *Sci. Data* 7, 1-18. <https://doi.org/10.1038/s41597-020-0479-6>

- Thank you for pointing out geomorpho90m. We agree that this would be an interesting dataset to include in models for beta-diversity. This would be an interesting route to pursue in a follow up paper. However, to our alpha diversity models we are afraid the dataset would not add much, because our goal was not to develop the most accurate models for continental-scale alpha diversity but to evaluate whether spectral diversity was an important predictor.

Methods:

You constantly describe that you use imaging spectrometer data from NEON from airborne campaigns. However, you also mention airborne images in lines 252-255. Please provide more detailed information on the imaging spectrometer data (camera type, number of wavelengths, recording times).

- We acknowledge that more details are needed to follow the most important steps. This was also mentioned by the editor and other reviewers. We have updated the methods accordingly. We used the RGB airborne images, which have a resolution of 10 cm, to guide the masking of shaded and non-vegetated areas, which would have been more difficult based on the spectral images with 1 m resolution only. We include this information now in the methods:

P15 L 295): "We checked the validity of our thresholds using NEON's high resolution camera imagery (DPI.30010.001)³⁴ with 10 cm spatial resolution, collected simultaneously with the spectral data."

Likewise, what corrections were made to correct for the temporal variability of possibly different acquisition times?

- We now clarify that we used brightness normalization to account for brightness differences due to different acquisition times and flight directions:

P15 L303: "For each site, we extracted spectral reflectance values per plot, brightness-normalized all spectra³⁵ to account for illumination differences and performed a PCA with type I-scaling to reduce data dimensionality."

In your study you use many different vegetation types. The spectral NEON data are available with a spatial resolution of 1x1m. Please mention whether a BRDF correction of the spectral data was carried out and what influence this has or could have on your model results.

- According to the documentation for NEON's spectrometer data, the correction for the Bidirectional Reflectance Distribution Function (BRDF) is not currently implemented as part of the standard atmospheric correction processing for the NEON imaging. This is because further work needs to be done to evaluate what BRDF scheme is appropriate for NEON data. We acknowledge that data quality might improve by using BRDF correction, but found it reasonable to use the data provided by NEON as is, because we think this is what most users would do. We clarify that ATCOR was used for spectral processing but without BRDF correction in the Methods section:

P14 L277: "Image processing is done by NEON and based on the industry standard ATCOR³¹ without correcting for BRDF (bidirectional reflectance distribution function) effects. Details can be found by following the dataset's DOI³⁰ and scrolling down to the documents under "Collection and Processing - Documentation"."

Your study did not describe how the LAI was calculated. Please explain this again.

- The LAI product we used in this study is developed by NEON. Details on the algorithm can be found on the product website under "Documentation" by following the link to the product DOI. We cite the DOI in the Methods. And we have updated the Methods to clarify how to access additional information for NEON products at two instances in the Methods:

P17 L358: "Environmental variables included ... and mean LAI calculated from NEON's spectrometer LAI mosaic data⁴⁸."

P14 L279: “Details can be found by following the dataset’s DOI³⁰ and scrolling down to the documents under “Collection and Processing - Documentation.”

P15 L314: “Details regarding the sampling design for generating these datasets can be found by following the datasets’ DOIs^{36,37} and scrolling down to the documents under “Collection and Processing - Documentation”.”

NEON uses the ATCOR output for their LAI product. From the documentation: “LAI, defined as the ratio of upper leaf surface area to ground area (for broadleaf canopies), or projected conifer needle surface area to ground area (for coniferous plants) for a given unit area, was calculated through ATCOR (a standard algorithm) using the Soil Adjusted Vegetation Index (SAVI) as input.”

Further recommendations:

As you discuss the issue of transferability and scaling of imaging spectrometer RS data, I do not understand why you do not include available spaceborne hyperspectral RS data in your analysis such as DESIS or PRISMA?

- We agree, testing our results using spaceborne hyperspectral RS data is critical and we hope to do this in a follow-up paper. Our main goal for this manuscript was testing the degree to which spectral diversity can predict plant diversity across different biomes and ecosystems, so it made sense to use the NEON AOP data as a first test of that hypothesis before going to spaceborne data. Ground data (plant inventories) acquired in a standardized way are critical for our study, and NEON provides currently the best data to follow our approach. While using spaceborne data is a next step, there are a number of additional challenges that would need to be considered. For example, the spatial accuracy of the DESIS data is 300-500 m, which would (without ground control points) have made it impossible to reliably interpret our results. Using NEON’s AOP data present in a way a “best case scenario”. In our opinion it is valuable to test the spectral diversity-plant diversity relationship using such “optimal” datasets as a first step, but we agree with the Reviewer that the second step would be using satellite data for sure.

We thank the Reviewer very much for reviewing our manuscript.

Reviewer #3 (Remarks to the Author):

Regarding the manuscript titled "Plant biodiversity across biomes captured by spectroscopy":

I recommend that this manuscript be published in this journal for the following reasons:

1. The findings that spectral diversity tracks taxonomic biodiversity, both alpha and beta, are extremely important.
2. Given the wide range of environmental conditions in the NEON sites, these results confirm expectation regarding expression of biodiversity in spectral signals better than has previously been done. As such, these results will serve well to lend credence to the explosion of spectroscopy-based diversity assessments that will take place as several satellites launch in this decade.
3. I also appreciate that the article is well-written (the Methods can use a little work, more on this below) and easy to follow, and the standardization in the NEON network of study sites leaves little room for complaint as far as soundness of the results.

I have very few comments about the main text...

- We thank the Reviewer and very much appreciate the positive assessment. For our responses to comments and suggestions please see below.

Line 89 - "plot-level spectra" is maybe a little too vague even if this is described later in the Methods, can you elaborate just a little to help the read understand.

- Certainly. We updated the sentence to specify that these are mean spectra per plot (L97). We also clarify "plot-level spectra" and how they are used for beta-diversity assessments in the Introduction:

P5 L97: "Testing the degree of correspondence between **ordinations of mean** plot-level spectra and plant inventories revealed significant covariance for 23 out of the 30 NEON sites with on average 47% of the total variation in plant inventories explained by spectra (Fig. 2b, Extended Data Table 3)."

P4 L75: "For investigating the spectral-diversity—plant-diversity relationship at the beta-scale, we used the same species inventories (beta-diversity metrics calculated from percent cover per species and plot) and plot-level spectral data (spectral variance among the mean spectra of 20 m x 20 m research plots)."

Line 122 - Similarly for "spectral alpha-diversity metrics", I think it would help if you briefly describe such a metric here.

- We thank the Reviewer for catching this. We changed the sentence so that it reads:

P6 L133: "In the case of the NEON imagery, with 1 m × 1 m pixels, spectral alpha-diversity **predicted plant alpha-diversity** best in forests with closed canopies (LAI ≥ 1), consisting of mature trees (crown diameter ≥ 2 m, Extended Data Fig. 3). "

"... spectral alpha-diversity metrics" didn't make sense here. We updated the introduction to briefly explain how spectral alpha and beta-diversity were calculated. And also updated the methods to include a short description of spectral diversity calculations instead of only citing the original paper by Laliberté et al. 2020.

P4 L72: "For investigating the spectral-diversity—plant-diversity relationship at the alpha-scale, we used plot-level species inventories (alpha-diversity metrics calculated from percent cover per species and plot) and pixel-level spectral data (spectral variance among the 1 m x 1 m image pixels per plot¹⁶). For investigating the spectral-diversity—plant-diversity relationship at the beta-scale, we used the same species inventories (beta-diversity metrics calculated from percent cover per species and plot) and plot-level spectral data (spectral variance among the mean spectra of 20 m x 20 m research plots)."

P15 L298: "Spectral diversity calculations followed the methodology published in Laliberté, Schweiger and Legendre (2020)¹⁶. We selected spectral alpha-diversity (SD_a) *sensu* Laliberté, Schweiger and Legendre (2020)¹⁶ over other existing spectral alpha-diversity metrics because it measures spectral variance, which we felt more closely reflected abundance-weighted taxonomic diversity measures. For all analyses, we used NEON's distributed base plots measuring 20 m x 20 m, as these are the plots within which plant inventories are conducted. For each site, we extracted spectral reflectance values per plot, brightness-normalized all spectra³⁵ to account for illumination differences and performed a PCA with type I-scaling to reduce data dimensionality. We selected PCs until more than 95% of the total spectral variation was explained. Spectral diversity was calculated as the total variance among the PC values of all pixels corrected by the number of pixels."

The methods section was a little less clearly written, and it left me with a few more questions, though nothing that

can't be fixed with some minor edits...

- We thank the Reviewer for pointing these issues out to us. A need to clarify the methods was also noted by the other reviewers. We carefully edited them and are confident that they read better now.

General

A really helpful addition to the Methods would be a brief description of the plot structure of a NEON site. I'm not entirely sure what the difference between an "inventory" plot and a "focal" plot.

- Actually, there is no difference between inventory and focal plots. Both refer to the plots we used in our study, which are the ones in which plant inventories were conducted. These are in fact NEON's distributed base plots. To prevent any confusion, we now explicitly state that we are using NEON's base plots for all analyses (L302). Throughout the rest of the manuscript, we call the base plots simply "plots". We think explaining the entire plot structure at a NEON site, which includes mammal plots, tick plots, soil plots etc., is not necessary in our case, since all we are interested in are the base plots.

P15 L302: "For all analyses, we used NEON's distributed base plots measuring 20 m x 20 m, as these are the plots within which plant inventories are conducted."

There are a few misspelled words (or rather correctly spelled, but wrong endings like "removed" instead of "remove" L261) and fairly inconsistent use of commas in this section that made it so I had to reread a few passages. It would be good to scan over this again fixing the words and perhaps breaking up long sentences to reduce comma use.

- We went through the methods section again and did our best to simplify sentences. We are confident that we caught all mistakes.

Line 255 - Good to state the spectral resolution here, 5 nm band spacing?

- We now state the spectral resolution the updated methods section:

P14 L276: "Spectral data comprised 426 bands, spanning the spectral region from ~380 nm to ~2510 nm at 5 nm band spacing."

Line 261 - Can you describe the NIR shade mask a bit more, starting with which band(s) and how are they combined?

- We clarify in the updated methods section that we used mean reflectance (ranging from 0-1) between 752 nm–1048 nm:

P14 L284: "We applied an NDVI (normalized difference vegetation index)-mask³² to exclude non-photosynthetically active vegetation and a near-infrared (NIR) shade-mask³³ calculated as mean reflectance (ranging from 0-1) between 752 nm and 1048 nm."

Line 266 - Is NIR > 0.18 here 18% reflectance?

- This is correct. 18% would be the mean reflectance over the NIR region specified above. We clarify that reflectance ranges from 0-1 in our case (L284, see above). We also realized that NIR, which is the abbreviation

for near-infrared, makes the description of thresholds unclear. We updated the section using “mean NIR reflectance ≥ 0.18 ” etc. (L378).

P14 L291: “As NIR **shade-mask** thresholds we chose **mean NIR reflectance ≥ 0.18** for plots dominated by forest, **mean NIR reflectance ≥ 0.2** for plots dominated by shrubland, and **mean NIR reflectance ≥ 0.22** for plots dominated by grassland.”

Line 268 - It sounds like the PCA was done across `_all_` spectra across all sites as well, and not by plot or some other subgrouping. Maybe just specify here if this is true.

- The PCA was applied across all spectra per site (but not across all sites). We clarify this in the updated version:

P15 L303: “For each site, we extracted spectral reflectance values per plot, brightness-normalized all spectra³⁵ to account for illumination differences and performed a PCA with type I-scaling to reduce data dimensionality.”

Line 270 - Can you briefly describe the spectral diversity methodology of Laliberté et al. 2020 here? It is fairly important to the results, and I don't think a reader should have to look this up.

- Of course. This was also pointed out by other reviewers. The section now reads:

P15 L298: “Spectral diversity calculations followed the methodology published in Laliberté, Schweiger and Legendre (2020)¹⁶. We selected spectral alpha-diversity (SD_α) *sensu* Laliberté, Schweiger and Legendre (2020)¹⁶ over other existing spectral alpha-diversity metrics because it measures spectral variance, which we felt more closely reflected abundance-weighted taxonomic diversity measures. For all analyses, we used NEON's distributed base plots measuring 20 m x 20 m, as these are the plots within which plant inventories are conducted. For each site, we extracted spectral reflectance values per plot, brightness-normalized all spectra³⁵ to account for illumination differences and performed a PCA with type I-scaling to reduce data dimensionality. We selected PCs until more than 95% of the total spectral variation was explained. Spectral diversity was calculated as the total variance among the PC values of all pixels corrected by the number of pixels.”

Line 284 - I'm not sure what the last clause of this sentence means. Maybe explain how shrub data are measured differently and why such a scaling needs to be done.

- In addition to the tree data (“woody plant vegetation structure”) we only used one inventory dataset (“plant presence and percent cover”), which includes shrubs. In short, the reason for scaling the inventory data is that it only includes plants $< 3\text{m}$ even when trees are present. For trees we have location, height and diameter information available, which we used to calculate tree cover per species. We then scaled the percent cover data for plants $< 3\text{m}$ to the area per plot that was not covered by trees. We have updated this section in methods and include an illustration (Extended Data Fig. 5) for clarity:

P15, L311: “We calculated taxonomic diversity metrics using **two datasets: i) percent cover of herbaceous and understory plants (i.e., plants $< 3\text{m}$)** from NEON's plant presence and percent cover data (DP1.10058.001)³⁶ and **ii) tree metrics, including identity, location, crown diameter and height, from NEON's woody plant vegetation structure data (DP1.10098.001)**³⁷.”

P16, L318: “Most plots and sites used in our study contained individuals $\geq 3\text{m}$ (trees, tall shrubs). However, for individuals $\geq 3\text{m}$ NEON's plant presence and percent cover (DP1.10058.001)³⁶ data does not report crown cover, but only basal diameter cover (when there is no vegetative growth $< 3\text{m}$) or percent cover of foliage along the stem $< 3\text{m}$ (when there is vegetative growth $< 3\text{m}$). For matching spectral data and plant inventories it is, however, important to scale inventories to vegetation cover as seen from above. We thus mapped all trees

recorded in the woody plant vegetation structure³⁷ data based on their location, crown diameter and height³⁸ (Extended Data Fig. 5).”

P16, L325: ”For each plot, we calculated crown cover per tree as seen from above taking overlapping crowns into account by ranking the trees according to their height. We then scaled plant cover < 3 m to the area per plot not covered by tree crowns and combined both, tree cover and scaled herbaceous/understory inventories. .”

Extended Data Fig. 5. Scaling plant inventories to vegetation cover as seen from above. Similar to this example of plot 1 (delineated by the blue square) at the Abby Road site, we mapped all trees per plot based on their location, height and crown diameter. We calculated the area within the plot covered by each tree species, and scaled percent cover of herbaceous and understory plants to the area within the plot not covered by trees (white area within the blue square).

We thank the Reviewer very much for the thoughtful review.

Reviewer comments, second round

Reviewer #1 (Remarks to the Author):

I think the authors have done a really good job of responding to all the various reviewer comments. They have made several positive modifications to what was already an excellent paper.

This is important ground-breaking research, the results are clear, the methodology is sound and the paper is well written. I would fully recommend that it is published in Nature Communications.

Reviewer #2 (Remarks to the Author):

Manuscript # NCOMMS-22-52204-T

I have looked at the response to my comments. The authors have implemented my comments very well and have given very good and comprehensible reasons why specific comments have not been implemented in the paper.

I therefore support the acceptance of the paper in the present form.

The paper should be accepted in that present form.